# Classification and Disease Localization in Histopathology Using Only Global Labels: A Weakly-Supervised Approach

## Abstract

Analysis of histopathology slides is a critical step for many diagnoses, and in particular in oncology where it defines the gold standard. In the case of digital histopathological analysis, highly trained pathologists must review vast whole-slide-images of extreme digital resolution ($100,000^2$ pixels) across multiple zoom levels in order to locate abnormal regions of cells, or in some cases single cells, out of millions. The application of deep learning to this problem is hampered not only by small sample sizes, as typical datasets contain only a few hundred samples, but also by the generation of ground-truth localized annotations for training interpretable classification and segmentation models. We propose a method for disease localization in the context of weakly supervised learning, where only image-level labels are available during training. Even without pixel-level annotations, we are able to demonstrate performance comparable with models trained with strong annotations on the Camelyon-16 lymph node metastases detection challenge. We accomplish this through the use of pre-trained deep convolutional networks, feature embedding, as well as learning via top instances and negative evidence, a multiple instance learning technique from the field of semantic segmentation and object detection.

## 1 Introduction

Histopathological image analysis (HIA) is a critical element of diagnosis in many areas of medicine, and especially in oncology, where it defines the gold standard metric. Recent works have sought to leverage modern developments in machine learning (ML) to aid pathologists in disease detection tasks, but the majority of these techniques require localized annotation masks as training data. These annotations are even more costly to obtain than the original diagnosis, as pathologists must spend time to assemble pixel-by-pixel segmentation maps of diseased tissue at extreme resolution, thus HIA datasets with annotations are very limited in size. Additionally, such localized annotations may not be available when facing new problems in HIA, such as new disease subtybe classification, prognosis estimation, or drug response prediction. Thus, the critical question for HIA is: can one design a learning architecture which achieves accurate classification with no additional localized annotation? A successful technique would be able train algorithms to assist pathologists during analysis, and could also be used to identify previously unknown structures and regions of interest.

Indeed, while histopathology is the gold standard diagnostic in oncology, it is extremely costly, requiring many hours of focus from pathologists to make a single diagnosis (Litjens et al., 2016; Weaver, 2010). Additionally, as correct diagnosis for certain diseases requires pathologists to identify a few cells out of millions, these tasks are akin to "finding a needle in a haystack." Hard numbers on diagnostic error rates in histopathology are difficult to obtain, being dependent upon the disease and tissue in question as well as self-reporting by pathologists of diagnostic errors. However, as reported in the review of Santana & Ferreira (2017), false negatives in cancer diagnosis can lead not only to catastrophic consequences for the patient, but also to incredible financial risk to the pathologist. Any tool which can aid pathologists to focus their attention and effort to the must suspect regions can help reduce false-negatives and improve patient outcomes through more accurate diagnoses (Djuric et al., 2017). Medical researchers have looked to computer-aided diagnosis for decades, but the lack of computational resources and data have prevented wide-spread implementa-

tion and usage of such tools (Gurcan et al., 2009). Since the advent of automated digital WSI capture in the 1990s, researchers have sought approaches for easing the pathologist's workload and improve patient outcomes through image processing algorithms (Gurcan et al., 2009; Litjens et al., 2017). Rather than predicting final diagnosis, many of these procedures focused instead on segmentation, either for cell-counting, or for the detection of suspect regions in the WSI. Historical methods have focused on the use of hand-crafted texture or morphological (Demir & Yener, 2005) features used in conjunction with unsupervised techniques such as K-means clustering or other dimensionality reduction techniques prior to classification via k-Nearest Neighbor or a support vector machine.

Over the past decade, fruitful developments in deep learning (LeCunn et al., 2015) have lead to an explosion of research into the automation of image processing tasks. While the application of such advanced ML techniques to image tasks has been successful for many consumer applications, the adoption of such approaches within the field of medical imaging has been more gradual. However, these techniques demonstrate remarkable promise in the field of HIA. Specifically, in digital pathology with whole-slide-imaging (WSI) (Yagi & Gilbertson, 2005; Snead et al., 2016), highly trained and skilled pathologists review digitally captured microscopy images from prepared and stained tissue samples in order to make diagnoses. Digital WSI are massive datasets, consisting of images captured at multiple zoom levels. At the greatest magnification levels, a WSI may have a digital resolution upwards of 100 thousand pixels in both dimensions. However, since localized annotations are very difficult to obtain, datasets may only contain WSI-level diagnosis labels, falling into the category of weakly-supervised learning.

The use of DCNNs was first proposed for HIA in Cireşan et al. (2013), where the authors were able to train a model for mitosis detection in H&E stained images. A similar technique was applied for WSI for the detection of invasive ductal carcinoma in Cruz-Roa et al. (2014). These approaches demonstrated the usefulness of learned features as an effective replacement for hand-crafted image features. It is possible to train deep architectures from scratch for the classification of tile images (Wang et al., 2016; Hou et al., 2016). However, training such DCNN architectures can be extremely resource intensive. For this reason, many recent approaches applying DCNNs to HIA make use of large pre-trained networks to act as rich feature extractors for tiles (Källén et al., 2016; Kim et al., 2016; Litjens et al., 2016; Xu et al., 2017; Song et al., 2017). Such approaches have found success as aggregation of rich representations from pre-trained DCNNs has proven to be quite effective, even without from-scratch training on WSI tiles.

In this paper, we propose CHOWDER[1], an approach for the interpretable prediction of general localized diseases in WSI with only weak, whole-image disease labels and without any additional expert-produced localized annotations, i.e. per-pixel segmentation maps, of diseased areas within the WSI. To accomplish this, we modify an existing architecture from the field of multiple instance learning and object region detection (Durand et al., 2016) to WSI diagnosis prediction. By modifying the pre-trained DCNN model (He et al., 2016), introducing an additional set of fully-connected layers for context-aware classification from tile instances, developing a random tile sampling scheme for efficient training over massive WSI, and enforcing a strict set of regualrizations, we are able to demonstrate performance equivalent to the best human pathologists (Bejnordi et al., 2017). Notably, while the approach we propose makes use of a pre-trained DCNN as a feature extractor, the entire procedure is a true end-to-end classification technique, and therefore the transferred pre-trained layers can be fine-tuned to the context of H&E WSI. We demonstrate, using only whole-slide labels, performance comparable to top-10 ranked methods trained with strong, pixel-level labels on the Camelyon-16 challenge dataset, while also producing disease segmentation that closely matches ground-truth annotations. We also present results for diagnosis prediction on WSI obtained from The Cancer Genome Atlas (TCGA), where strong annotations are not available and diseases may not be strongly localized within the tissue sample.

## 2 LEARNING WITHOUT LOCAL ANNOTATIONS

While approaches using localized annotations have shown promise for HIA, they fail to address the cost associated with the acquisition of hand-labeled datasets, as in each case these methods require access to pixel-level labels. As shown with ImageNet (Deng et al., 2009), access to data drives innovation, however for HIA hand-labeled segmentation maps are costly to produce, often subject

---

[1]Classification of HistOpathology with Weak supervision via Deep fEature aggRegation

to missed diseased areas, and cannot scale to the size of datasets required for truly effective deep learning. Because of these considerations, HIA is uniquely suited to the *weakly supervised learning* (WSL) setting.

Here, we define the WSL task for HIA to be the identification of suspect regions of WSI when the training data only contains image-wide labels of diagnoses made by expert pathologists. Since WSI are often digitally processed in small patches, or tiles, the aggregation of these tiles into groups with a single label (e.g. "healthy", "cancer present") can be used within the framework of *multiple instance learning* (MIL) (Dietterich et al., 1997; Amores, 2013; Xu et al., 2014). In MIL for binary classification, one often makes the standard multi-instance (SMI) assumption: a bag is classified as positive iff at least *one* instance (here, a tile) in the bag is labelled positive. The goal is to take the bag-level labels and learn a set of instance-level rules for the classification of single instances. In the case of HIA, learning such rules provides the ability to infer localized regions of abnormal cells within the large-scale WSI.

In the recent work of Hou et al. (2016) for WSI classification in the WSL setting, the authors propose an EM-based method to identify discriminative patches in high resolution images automatically during patch-level CNN training. They also introduced a decision level fusion method for HIA, which is more robust than max-pooling and can be thought of as a Count-based Multiple Instance (CMI) learning method with two-level learning. While this approach was shown to be effective in the case of glioma classification and obtains the best result, it only slightly outperforms much simpler approaches presented in (Hou et al., 2016), but at much greater computational cost.

In the case of natural images, the WELDON and WILDCAT techniques of Durand et al. (2016) and Durand et al. (2017), respectively, demonstrated state-of-the-art performance for object detection and localization for WSL with image-wide labels. In the case of WELDON, the authors propose an end-to-end trainable CNN model based on MIL learning with top instances (Li & Vasconcelos, 2015) as well as negative evidence, relaxing the SMI assumption. Specifically, in the case of semantic segmentation, Li & Vasconcelos (2015) argue that a target concept might not exist just at the subregion level, but that the proportion of positive and negative samples in a bag have a larger effect in the determination of label assignment. This argument also holds for the case of HIA, where pathologist diagnosis arises from a synthesis of observations across multiple resolution levels as well as the relative abundance of diseased cells. In Sec. 2.3, we will detail our proposed approach which makes a number of improvements on the framework of Durand et al. (2016), adapting it to the context of large-scale WSI for HIA.

## 2.1 WSI Pre-Processing

**Tissue Detection.**    As seen in Fig. 1, large regions of a WSI may contain no tissue at all, and are therefore not useful for training and inference. To extract only tiles with content relevant to the task, we use the same approach as Wang et al. (2016), namely, Otsu's method (Otsu, 1979) applied to the hue and saturation channels of the image after transformation into the HSV color space to produce two masks which are then combined to produce the final tissue segmentation. Subsequently, only tiles within the foreground segmentation are extracted for training and inference.

**Color Normalization.**    According to Ciompi et al. (2017), stain normalization is an important step in HIA since the result of the H&E staining procedure can vary greatly between any two slides. We utilize a simple histogram equalization algorithm consisting of left-shifting RGB channels and subsequently rescaling them to $[0, 255]$, as proposed in Nikitenko et al. (2008). In this work, we place a particular emphasis on the tile aggregation method rather than color normalization, so we did not make use of more advanced color normalization algorithms, such as Khan et al. (2014).

**Tiling.**    The tiling step is necessary in histopathology analysis. Indeed, due to the large size of the WSI, it is computationally intractable to process the slide in its entirety. For example, on the highest resolution zoom level, denoted as *scale 0*, for a fixed grid of non-overlapping tiles, a WSI may possess more than $200,000$ tiles of $224 \times 224$ pixels. Because of the computational burden associated with processing the set of all possible tiles, we instead turn to a uniform random sampling from the space of possible tiles. Additionally, due to the large scale nature of WSI datasets, the computational burden associated with sampling potentially overlapping tiles from arbitrary locations is a prohibitive cost for batch construction during training.

Instead, we propose that all tiles from the non-overlapping grid should be processed and stored to disk prior to training. As the tissue structure does not exhibit any strong periodicity, we find that sampling tiles along a fixed grid without overlapping provides a reasonably representative sampling while maximizing the total sampled area.

Given a target scale $\ell \in \{0, 1, \ldots, L\}$, we denote the number of possible tiles in WSI indexed by $i \in \{1, 2, \ldots, N\}$ as $M_{i,\ell}^{\mathrm{T}}$. The number of tiles sampled for training or inference is denoted by $M_{i,\ell}^{\mathrm{S}}$ and is chosen according to

$$M_{i,\ell}^{\mathrm{S}} = \min\left( M_{i,\ell}^{\mathrm{T}}, \ \max\left( M_{\min}^{\mathrm{T}}, \ \frac{1}{2} \cdot \bar{M}_{\ell}^{\mathrm{T}} \right) \right), \tag{1}$$

where $\bar{M}_{\ell}^{\mathrm{T}} = \frac{1}{N} \sum_i M_{i,\ell}^{\mathrm{T}}$ is the empirical average of the number of tiles at scale $\ell$ over the entire set of training data.

**Feature Extraction.** We make use of the ResNet-50 (He et al., 2016) architecture trained on the ImageNet natural image dataset. In empirical comparisons between VGG or Inception architectures, we have found that the ResNet architecture provides features more well suited for HIA. Additionally, the ResNet architecture is provided at a variety of depths (ResNet-101, ResNet-152). However, we found that ResNet-50 provides the best balance between the computational burden of forward inference and richness of representation for HIA.

In our approach, for every tile we use the values of the ResNet-50 pre-output layer, a set of $P = 2048$ floating point values, as the feature vector for the tile. Since the fixed input resolution for ResNet-50 is $224 \times 224$ pixels, we set the resolution for the tiles extracted from the WSI to the same pixel resolution at every scale $\ell$.

## 2.2 Baseline method

Given a WSI, extracting tile-level features produces a bag of feature vectors which one attempts to use for classification against the known image-wide label. The dimension of these local descriptors is $M^{\mathrm{S}} \times P$, where $P$ is the number of features output from the pre-trained image DCNN and $M^{\mathrm{S}}$ is the number of sampled tiles. Approaches such as Bag-of-visual-words (BoVW) or VLAD (Jégou et al., 2010) could be chosen as a baseline aggregation method to generate a single image-wide descriptor of size $P \times 1$, but would require a huge computational power given the dimensionality of the input. Instead, we will try two common approaches for the aggregation of local features, specifically, the `MaxPool` and `MeanPool` and subsequently apply a classifier on the aggregated features. After applying these pooling methods over the axis of tile indices, one obtains a single feature descriptor for the whole image. Other pooling approaches have been used in the context of HIA, including Fisher vector encodings (Song et al., 2017) and $p-$norm pooling (Xu et al., 2017). However, as the reported effect of these aggregations is quite small, we don't consider these approaches when constructing our baseline approach.

After aggregation, a classifier can be trained to produce the desired diagnosis labels given the global WSI aggregated descriptor. For our baseline method, we use a logistic regression for this final prediction layer of the model. We present a description of the baseline approach in Fig. 1.

## 2.3 Chowder Method

In experimentation, we observe that the baseline approach of the previous section works well for *diffuse* disease, which is evidenced in the results of Table 1 for `TCGA-Lung`. Here, *diffuse* implies that the number of disease-containing tiles, pertinent to the diagnosis label, are roughly proportional to the number of tiles containing healthy tissue. However, if one applies the same approach to different WSI datasets, such as `Camelyon-16`, the performance significantly degrades. In the case of `Camelyon-16`, the diseased regions of most of the slides are highly localized, restricted to a very small area within the WSI. When presented with such imbalanced bags, simple aggregation approaches for global slide descriptors will overwhelm the features of the disease-containing tiles.

Instead, we propose an adaptation and improvement of the WELDON method (Durand et al., 2016) designed for histopathology images analysis. As in their approach, rather than creating a global

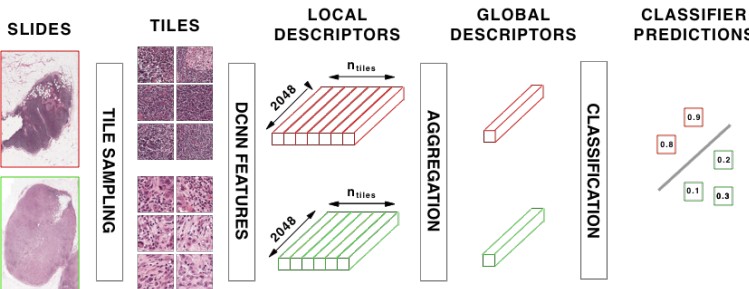

Figure 1: Description of the BASELINE approach for WSI classification via aggregation of tile-level features into global slide descriptors.

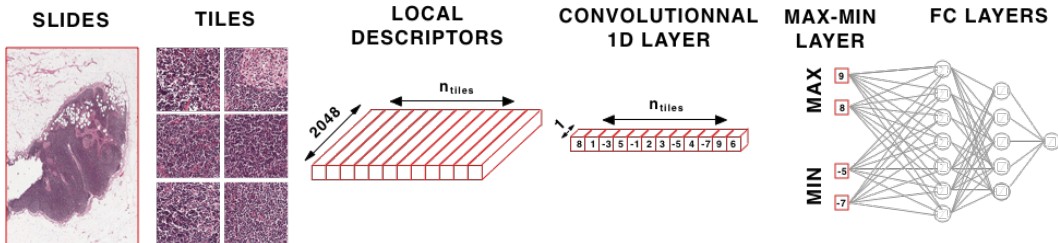

Figure 2: Description of the CHOWDER architecture (for $R = 2$) for WSI classification via MLP on operating on top positive and negative instances shown for a single sample mini-batch sample.

slide descriptor by aggregating all tile features, instead a MIL approach is used that combines both top-instances as well as negative evidence. A visual description of approach is given in Fig. 2.

**Feature Embedding.** First, a set of one-dimensional embeddings for the $P = 2048$ ResNet-50 features are calcualted via $J$ one-dimensional convolutional layers strided across the tile index axis. For tile $t$ with features $\mathbf{k}_t$, the embedding according to kernel $j$ is calculated as $e_{j,t} = \langle \mathbf{w}_j, \mathbf{k}_t \rangle$. Notably, the kernels $\mathbf{w}_j$ have dimensionality $P$. This one-dimensional convolution is, in essence, a shortcut for enforcing a fully-connected layer with tied weights across tiles, i.e. the same embedding for every tile (Durand et al., 2016). In our experiments, we found that the use of a single embedding, $J = 1$, is an appropriate choice for WSI datasets when the number of available slides is small ($< 1000$). In this case, choosing $J > 1$ will decrease training error, but will *increase* generalization error. Avoiding overtraining and ensuring model generality remains a major challenge for the application of WSL to WSI datasets.

**Top Instances and Negative Evidence.** After feature embedding, we now have a $M_{\ell,i}^{\mathrm{S}} \times 1$ vector of local tile-level (*instance*) descriptors. As in (Durand et al., 2016), these instance descriptors are sorted by value. Of these sorted embedding values, only the top and bottom $R$ entries are retained, resulting in a tensor of $2R \times 1$ entries to use for diagnosis classification. This can be easily accomplished through a `MinMax` layer on the output of the one-dimensional convolution layer. The purpose of this layer is to take not only the top instances region but also the negative evidences, that is the region which best support the absence of the class. During training, the back-propagation runs only through the selected tiles, positive and negative evidences. When applied to WSI, the `MinMax` serves as a powerful tile selection procedure.

**Multi-layer Perceptron (MLP) Classifier.** In the WELDON architecture, the last layer consists of a sum applied over the $2R \times 1$ output from the `MinMax` layer. However, we find that this approach can be improved for WSI classification. We investigate the possibility of a richer interactions between the top and bottom instances by instead using an MLP as the final classifier. In our implementation of CHOWDER, we use an MLP with two fully connected layers of 200 and 100 neurons with sigmoid activations.

# 3 EXPERIMENTAL RESULTS

## 3.1 TRAINING DETAILS

First, for pre-processing, we fix a single tile scale for all methods and datasets. We chose a fixed zoom level of 0.5 $\mu$m/pixel, which corresponds to $\ell = 0$ for slides scanned at 20x magnification, or $\ell = 1$ slides scanned at 40x magnification. Next, since WSI datasets often only contain a few hundred images, far from the millions images of ImageNet dataset, strong regularization required prevent over-fitting. We applied $\ell_2$-regularization of 0.5 on the convolutional feature embedding layer and dropout on the MLP with a rate of 0.5. However, these values may not be the global optimal, as we did not apply any hyper-parameter optimization to tune these values. To optimize the model parameters, we use Adam (Kingma & Ba, 2014) to minimize the binary cross-entropy loss over 30 epochs with a mini-batch size of 10 and with learning rate of 0.001.

To reduce variance and prevent over-fitting, we trained an ensemble of $E$ CHOWDER networks which only differ by their initial weights. The average of the predictions made by these $E$ networks establishes the final prediction. Although we set $E = 10$ for the results presented in Table 1, we used a larger ensemble of $E = 50$ with $R = 5$ to obtain the best possible model and compare our method to those presented in Table 2. We also use an ensemble of $E = 10$ when reporting the results for WELDON. As the training of one epoch requires about 30 seconds on our available hardware, the total training time for the ensemble took just over twelve hours. While the ResNet-50 features were extracted using a GPU for efficient feed-forward calculations, the CHOWDER network is trained on CPU in order to take advantage of larger system RAM sizes, compared to on-board GPU RAM. This allows us to store all the training tiles in memory to provide faster training compared to a GPU due to reduced transfer overhead.

## 3.2 TCGA

The public Cancer Genome Atlas (TCGA) provides approximately 11,000 tissue slides images of cancers of various organs[2]. For our first experiment, we selected 707 lung cancer WSIs (`TCGA-Lung`), which were downloaded in March 2017. Subsequently, a set of new lung slides have been added to TCGA, increasing the count of lung slides to 1,009. Along with the slides themselves, TCGA also provides labels representing the type of cancer present in each WSI. However, no local segmentation annotations of cancerous tissue regions are provided. The pre-processing step extracts 1,411,043 tiles and their corresponding representations from ResNet-50. The task of these experiments is then to predict which type of cancer is contained in each WSI: adenocarcinoma or squamous cell carcinoma. We evaluate the quality of the classification according to the area under the curve (AUC) of the receiver operating characteristic (ROC) curve generated using the raw output predictions.

As expected in the case of diffuse disease, the advantage provided by CHOWDER is slight as compared to the `MeanPool` baseline, as evidenced in Table 1. Additionally, as the full aggregation techniques work quite well in this setting, the value of $R$ does not seem to have a strong effect on the performance of CHOWDER as it increases to $R = 100$. In this setting of highly homogenous tissue content, we can expect that global aggregate descriptors are able to effectively separate the two classes of carcinoma.

## 3.3 CAMELYON-16

For our second experiment, we use the `Camelyon-16` challenge dataset[3], which consists of 400 WSIs taken from sentinel lymph nodes, which are either healthy or exhibit metastases of some form. In addition to the WSIs themselves, as well as their labeling (`healthy`, `contains-metastases`), a segmentation mask is provided for each WSI which represents an expert analysis on the location of metastases within the WSI. Human labeling of sentinel lymph node slides is known to be quite tedious, as noted in Litjens et al. (2016); Weaver (2010). Teams participating in the challenge had access to, and utilized, the ground-truth masks when training their diagnosis prediction and tumor localization models. For our approach, we set aside the masks of

---

[2]https://portal.gdc.cancer.gov/legacy-archive
[3]https://camelyon16.grand-challenge.org

metastasis locations and utilize only diagnosis labels. Furthermore, many participating teams developed a post-processing step, extracting handcrafted features from predicted metastasis maps to improve their segmentation. No post-processing is performed for the presented CHOWDER results, the score is computed directly from the raw output of the CHOWDER model.

The `Camelyon-16` dataset is evaluated on two different axes. First, the accuracy of the predicted label for each WSI in the test set is evaluated according to AUC. Second, the accuracy of metastasis localization is evaluated by comparing model outputs to the ground-truth expert annotations of metastasis location. This segmentation accuracy is measured according to the free ROC metric (FROC), which is the curve of metastasis detection sensitivity to the average number of also positives. As in the Camelyon challenge, we evaluate the FROC metric as the average detection sensitivity at the average false positive rates 0.25, 0.5, 1, 2, 4, and 8.

**Competition Split Bias.** We also conduct a set of experiments on `Camelyon-16` using random train-test cross-validation (CV) splits, respecting the same training set size as in the original competition split. We note distinct difference in AUC between the competition split and those obtained via random folds. This discrepancy is especially distinct for the `MeanPool` baseline, as reported in Table 1. We therefore note a distinct discrepancy in the data distribution between the competition test and training splits. Notably, using the `MeanPool` baseline architecture, we found that the competition train-test split can be predicted with an AUC of 0.75, however one only obtains an AUC of 0.55 when using random splits. Because this distribution mismatch in the competition split could produce misleading interpretations, we report 3-fold average CV results along with the results obtained on the competition split.

**Classification Performance.** In Table 1, we see the classification performance of our proposed CHOWDER method, for $E = 10$, as compared to both the baseline aggregation techniques, as well as the WELDON approach. In the case of WELDON, the final MLP is not used and instead a summing is applied to the `MinMax` layer. The value of $R$ retains the same meaning in both cases: the number of both high and low scoring tiles to pass on to the classification layers. We test a range of values $R$ for both WELDON and CHOWDER. We find that over all values of $R$, CHOWDER provides a significant advantage over both the baseline aggregation techniques as well as WELDON. We also note that the optimal performance can be obtained without using a large number of discriminative tiles, i.e. $R = 5$.

We also present in Table 2 our performance as compared to the public Camelyon leader boards for $E = 50$. In this case, we are able to obtain an effective $11^{\text{th}}$ place rank, *but without using any of the ground-truth disease segmentation maps*. This is a remarkable result, as the winning approach of Wang et al. (2016) required tile-level disease labels derived from expert-provided annotations in order to train a full 27-layer GoogLeNet (Szegedy et al., 2015) architecture for tumor prediction. We also show the ROC curve for this result in Fig. 3. Finally, we note that CHOWDER's performance on this task roughly is equivalent to the best-performing human pathologist, an AUC of 0.884 as reported by Bejnordi et al. (2017), and better than the average human pathologist performance, an AUC of 0.810. Notably, this human-level performance is achieved without human assistance during training, beyond the diagnosis labels themselves.

**Localization Performance.** Obtaining high performance in terms of whole slide classification is well and good, but it is not worth much without an interpretable result which can be used by pathologists to aid their diagnosis. For example, the `MeanPool` baseline aggregation approach provides no information during inference from which one could derive tumor locations in the WSI: all locality information is lost with the aggregation. With `MaxPool`, one at least retains some information via the tile locations which provide each maximum aggregate feature.

For CHOWDER, we propose the use of the full set of outputs from the convolutional feature embedding layer. These are then sorted and thresholded according to value $\tau$ such that tiles with an embedded value larger than $\tau$ are classified as diseased and those with lower values are classified as healthy. We show an example of disease localization produced by CHOWDER in Fig. 4. Here, we see that CHOWDER is able to very accurately localize the tumorous region in the WSI even though it has only been trained using global slide-wide labels and without any local annotations. While some potential false detections occur outside of the tumor region, we see that the strongest

| | AUC | |
|---|---|---|
| **Method** | *CV* | *Competition* |
| *BASELINE* | | |
| `MaxPool` | 0.749 | 0.655 |
| `MeanPool` | 0.802 | 0.530 |
| *WELDON* | | |
| $R = 1$ | 0.782 | 0.765 |
| $R = 10$ | 0.832 | 0.670 |
| $R = 100$ | 0.809 | 0.600 |
| $R = 300$ | 0.761 | 0.573 |
| *CHOWDER* | | |
| $R = 1$ | 0.809 | 0.821 |
| $R = 5$ | **0.903** | **0.858** |
| $R = 10$ | 0.900 | 0.843 |
| $R = 100$ | 0.870 | 0.775 |
| $R = 300$ | 0.837 | 0.652 |

| **Method** | **AUC** |
|---|---|
| *BASELINE* | |
| `MaxPool` | 0.860 |
| `MeanPool` | 0.903 |
| *CHOWDER* | |
| $R = 1$ | 0.900 |
| $R = 10$ | **0.915** |
| $R = 100$ | 0.909 |

Table 1: Classification (AUC) results for the `Camelyon-16` (**left**) and `TCGA-Lung` (**right**) datasets for CHOWDER, WELDON, and the baseline approach. For `Camelyon-16`, we present two scores, one for the fixed competition test split of 130 WSIs, and one for a cross-validated average over 3 folds (*CV*) on the 270 training WSIs. For `TCGA-Lung`, we present scores as a cross-validated average over 5 folds.

response occurs within the tumor region itself, and follows the border regions nicely. We present further localization results in Appendix A.

We also present FROC scores for CHOWDER in Table 2 as compared to the leader board results. Here, we obtain results comparable to the $18^{\text{th}}$ rank. However, this performance is incredibly significant as all other approaches were making use of tile-level classification in order to train their segmentation techniques. We also show the FROC curve in Fig. 3.

## 4 DISCUSSION

We have shown that using state-of-the-art techniques from MIL in computer vision, such as the top instance and negative evidence approach of (Durand et al., 2016), one can construct an effective technique for diagnosis prediction *and* disease location for WSI in histopathology without the need

| Rank | Team | AUC | Rank | Team | FROC |
|---|---|---|---|---|---|
| 1 | HMS & MIT | 0.9935 | 1 | HMS & MIT | 0.8074 |
| 2 | HMS-MGH | 0.9763 | 2 | HMS-MGH | 0.7600 |
| 3 | HMS-MGH | 0.9650 | 3 | HMS-MGH | 0.7289 |
| 4 | CUHK | 0.9415 | 4 | CUHK | 0.7030 |
| | . . . | | | . . . | |
| 9 | CUHK | 0.9056 | 16 | Osaka University | 0.3467 |
| 10 | DeepCare Inc. | 0.8833 | 17 | SIT | 0.3385 |
| | **CHOWDER (No Annotation)** | **0.8706** | | **CHOWDER (No Annotation)** | **0.3103** |
| 11 | Indep. DE | 0.8654 | 18 | Warwick-QU | 0.3052 |
| 12 | METU | 0.8642 | 19 | U. Munich (CAMP) | 0.2733 |
| | . . . | | | . . . | |
| 32 | Sorbonne LIB | 0.5561 | 32 | Mines Paris Tech | 0.0970 |

Table 2: Final leader boards for `Camelyon-16` competition. All competition methods had access to the full set of strong annotations for training their models. In contrast, our proposed approach only utilizes image-wide diagnosis levels and obtains comparable performance as top-10 methods.

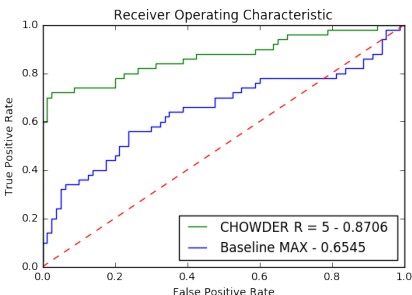 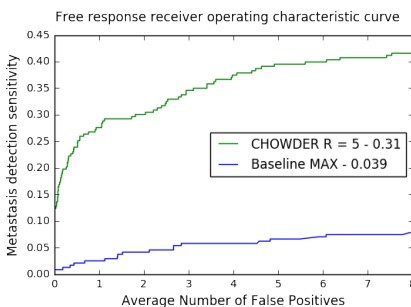

Figure 3: Performance curves for `Camelyon-16` dataset for both classification and segmentation tasks for the different tested approaches. **Left:** ROC curves for the classification task. **Right:** FROC curves for lesion detection task.

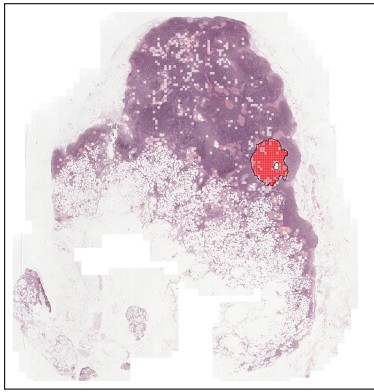 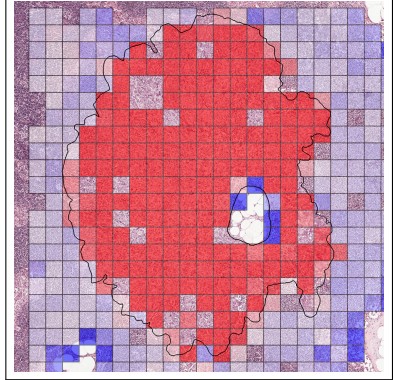

Figure 4: Visualization of metastasis detection on test image 27 of the `Camelyon-16` dataset using our proposed approach. **Left:** Full WSI at zoom level 6 with ground truth annotation of metastases shown via black border. Tiles with positive feature embeddings are colored from white to red according to their magnitude, with red representing the largest magnitude. **Right:** Detail of metastases at zoom level 2 overlaid with classification output of our proposed approach. Here, the output of all tested tiles are shown and colored according to their value, from blue to white to red, with blue representing the most negative values, and red the most positive. Tiles without color were not included when randomly selecting tiles for inference.

for expensive localized annotations produced by expert pathologists. By removing this requirement, we hope to accelerate the production of computer-assistance tools for pathologists to greatly improve the turn-around time in pathology labs and help surgeons and oncologists make rapid and effective patient care decisions. This also opens the way to tackle problems where expert pathologists may not know precisely where relevant tissue is located within the slide image, for instance for prognosis estimation or prediction of drug response tasks. The ability of our approach to discover associated regions of interest without prior localized annotations hence appears as a novel discovery approach for the field of pathology. Moreover, using the suggested localization from CHOWDER, one may considerably speed up the process of obtaining ground-truth localized annotations.

A number of improvements can be made in the CHOWDER method, especially in the production of disease localization maps. As presented, we use the raw values from convolutional embedding layer, which means that the resolution of the produced disease localization map is fixed to that of the sampled tiles. However, one could also sample overlapping tiles and then use a data fusion technique to generate a final localization map. Additionally, as a variety of annotations may be available, CHOWDER could be extended to the case of heterogenous annotation, e.g. some slides with expert-produced localized annotations and those with only whole-slide annotations.

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

## A    FURTHER RESULTS

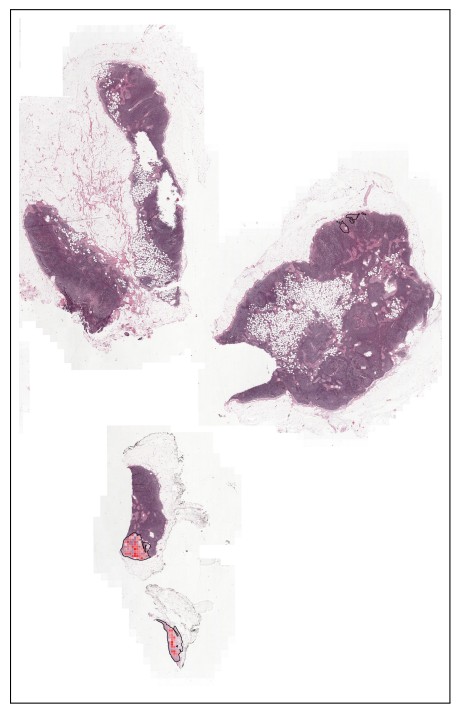

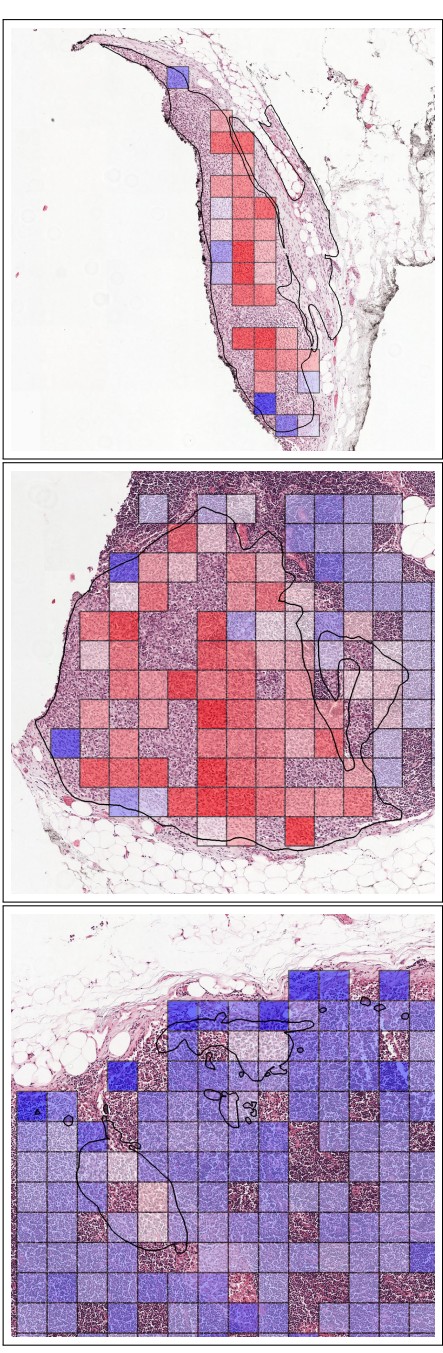

Figure 5: Visualization of metastasis detection on test image 2 of the `Camelyon-16` dataset using our proposed approach. **Left:** Full WSI at zoom level 6 with ground truth annotation of metastases shown via black border. Tiles with positive feature embeddings are colored from white to red according to their magnitude, with red representing the largest magnitude. **Right:** Detail of metastases at zoom level 2 overlaid with classification output of our proposed approach. Here, the output of all tested tiles are shown and colored according to their value, from blue to white to red, with blue representing the most negative values, and red the most positive. Tiles without color were not included when randomly selecting tiles for inference.

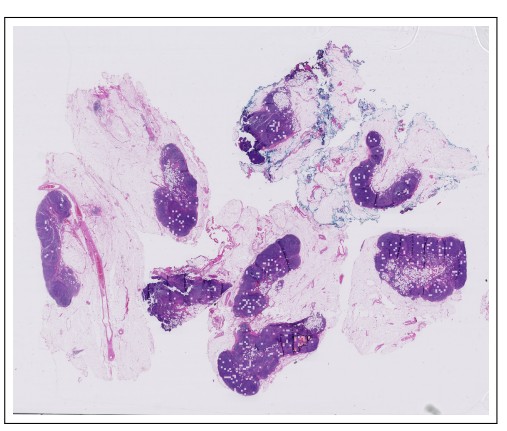 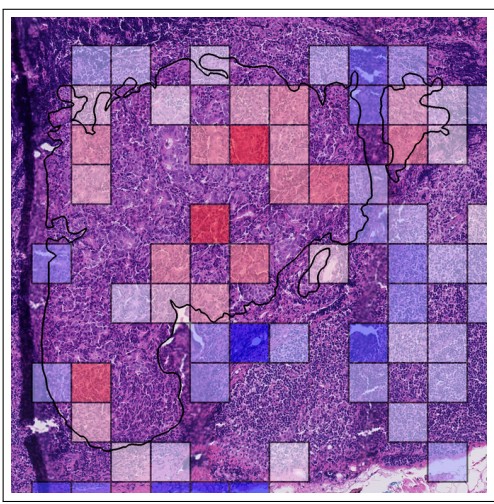

Figure 6: Visualization of metastasis detection on test image 92 of the `Camelyon-16` dataset using our proposed approach. **Left:** Full WSI at zoom level 6 with ground truth annotation of metastases shown via black border. Tiles with positive feature embeddings are colored from white to red according to their magnitude, with red representing the largest magnitude. **Right:** Detail of metastases at zoom level 2 overlaid with classification output of our proposed approach. Here, the output of all tested tiles are shown and colored according to their value, from blue to white to red, with blue representing the most negative values, and red the most positive. Tiles without color were not included when randomly selecting tiles for inference.

