# OpenReview forum: "Classification and Disease Localization in Histopathology Using Only Global Labels: A Weakly-Supervised Approach"
_ICLR.cc/2018/Conference — Reject_

### Official Review · AnonReviewer3 · 2017-11-20
**Interesting MIL approach, lacks technical depth for this conference**

**Rating:** 5
**Confidence:** 4

**Review:**

This paper describes a semi-supervised method to classify and segment WSI histological images that are only labeled at the whole image level. Images are tiled and tiles are sampled and encoded into a feature vector via a ResNET-50 pretrained on ImageNET. A 1D convolutional layer followed by a min-max layer and 2 fully connected layer compose the network. The conv layer produces a single value per tile. The min-max layer selects the R min and max values, which then enter the FC layers. A multi-instance (MIL) approach is used to train the model by backpropagating only instances that generate min and max values at the min-max layer. Experiments are run on 2 public datasets achieving potentially top performance. Potentially, because all other methods supposedly make use of segmentation labels of tumor, while this method only uses the whole image label.

Previous publications have used MIL training on tiles with only top-level labels [1,2] and this is essentially an incremental improvement on the MIL approach by using several instances (both min-negative and max-positive) instead of a single instance for backprop, as described in [3]. So, the main contribution here, is to adapt min-max MIL to the histology domain. Although the result are good and the method interesting, I think that the technical contribution is a bit thin for a ML conference and this paper may be a better fit for a medical imaging conference.

The paper is well written and easy to understand.



[1] Hou, L., Samaras, D., Kurc, T. M., Gao, Y., Davis, J. E., & Saltz, J. H. (2016). Patch-based convolutional neural network for whole slide tissue image classification. In Proceedings of the IEEE Conference on Computer Vision and Pattern Recognition (pp. 2424-2433).
[2]  Cosatto, E., Laquerre, P. F., Malon, C., Graf, H. P., Saito, A., Kiyuna, T., ... (2013). Automated gastric cancer diagnosis on H&E-stained sections; training a classifier on a large scale with multiple instance machine learning. Medical Imaging, 2. 2013.
[3] Durand, T., Thome, N., & Cord, M. (2016). Weldon: Weakly supervised learning of deep convolutional neural networks. In Proceedings of the IEEE Conference on Computer Vision and Pattern Recognition (pp. 4743-4752).

---

> ### Author Response · Authors · 2018-01-05
> **Response to Referee**
>
> We thank the referee for their time and effort in assessing our work. We hope that the
> discussion we provide in our general comments provides further justification
> for our work's presence at ICLR. Specifically, we note the significance of our
> architectural contributions, as well as our modifications to the training regime. We
> also detail how our system provides human-pathologist-level performance without being
> guided by detailed expert instruction on what structures lead to disease diagnoses.
> We believe that this significant advance in machine learning as applied to medical
> imaging, and to this gold standard oncology diagnostic in particular, will be of
> great interest to the general ICLR audience.
>
> >Previous publications have used MIL training on tiles with only top-level labels [1,2]
> >and this is essentially an incremental improvement on the MIL approach by using
> >several instances (both min-negative and max-positive) instead of a single instance
> >for backprop, as described in [3]. So, the main contribution here, is to adapt min-max
> >MIL to the histology domain. Although the result are good and the method interesting,
> >I think that the technical contribution is a bit thin for a ML conference and this paper
> >may be a better fit for a medical imaging conference.
>
> We thank the referee for their positive view of our method and results. We agree that
> the work we present is, as Reviewer1 noted, a "down-to-earth practical application,"
> however we do make novel architectural, process, and implementation contributions. While
> we do not provide a theory of MIL in the context of HIA, we note that many successful
> advances in our field have been made from an empirical, rather than theoretical,
> perspective. While there is no newly proposed loss, neuron non-linearity, or adaptive
> momentum scheme in our work, we do demonstrate the steps necessary to provide
> state-of-the-art performance for diagnosis prediction and disease localization without
> expert assistance beyond diagnosis labels. While these results would indeed be
> incredibly pertinent at a more medically focused venue, it is our strong belief that the audience
> of ICLR would greatly benefit both from our demonstration, as well as their introduction
> to a budding application area in great need of their technical expertise.
>
> >The paper is well written and easy to understand.
>
> We thank the referee for their comments and positive feedback on our presentation of our
> work.

---

### Official Review · AnonReviewer1 · 2017-12-02
**Down-to-earth practical application of DL in a medico-clinical context**

**Rating:** 6
**Confidence:** 3

**Review:**

This paper proposes a deep learning (DL) approach (pre-trained CNNs) to the analysis of histopathological images for disease localization.
It correctly identifies the problem that DL usually requires large image databases to provide competitive results, while annotated histopathological data repositories are costly to produce and not on that size scale.
It also correctly identifies that this is a daunting task for human medical experts and therefore one that could surely benefit from the use of automated methods like the ones proposed.

The study seems sound from a technical viewpoint to me and its contribution is incremental, as it builds on existing research, which is correctly identified.
Results are not always too impressive, but authors seem intent on making them useful for pathogists in practice (an intention that is always worth the effort).
I think the paper would benefit from a more explicit statement of its original contributions (against contextual published research)

Minor issues:
Revise typos (e.g. title of section 2)
Please revise list of references (right now a mess in terms of format, typos, incompleteness

---

> ### Author Response · Authors · 2018-01-05
> **Response to Referee**
>
> >The study seems sound from a technical viewpoint to me and its contribution is incremental,
> >as it builds on existing research, which is correctly identified.
> >Results are not always too impressive, but authors seem intent on making them useful for
> > pathogists in practice (an intention that is always worth the effort).
> >I think the paper would benefit from a more explicit statement of its original contributions
> >(against contextual published research)
>
> We thank the referee for their comments and their effort in assessing our work.
> With respect to our specific contributions, we have added further clarifications to
>  the text (as noted in the paper modifications) to identify our contribution with
> respect to prior art.  Additionally, with respect to the significance of the results we
> present, we note that the performance reported in Table 1 represents the
> state-of-the-art for HIA classification using only WSI-wide labels. For further
> justifications on the significance of our work, we refer to our general comments
> on this subject.
>
> >Minor issues:
> >Revise typos (e.g. title of section 2)
>
> Thank you for pointing out this (rather embarrassing) typo! We have corrected this
> mistake along with others throughout the text.
>
> >Please revise list of references (right now a mess in terms of format, typos, incompleteness
>
> As noted in the general comments, we have revised the references to fit a common standard
> and have attempted to include all relevant citation details. We thank you for your
> attentiveness.

---

### Official Review · AnonReviewer2 · 2017-12-05
**Interesting application and results with incremental innovation on exististing work**

**Rating:** 5
**Confidence:** 3

**Review:**

The authors approach the task of labeling histology images with just a single global label, with promising results on two different data sets. This is of high relevance given the difficulty in obtaining expert annotated data. At the same time the key elements of the presented approach remain identical to those in a previous study, the main novelty is to replace the final step of the previous architecture (that averages across a vector) with a multiplayer perceptron.  As such I feel that this would be interesting to present if there is interest in the overall application (and results of the 2016 CVPR paper), but not necessarily as a novel contribution to MIL and histology image classification.

Comments to the authors:

* The intro starts from a very high clinical level. A introduction that points out specifics of the technical aspects of this application, the remaining technical challenges, and the contribution of this work might be appreciated by some of your readers.
* There is preprocessing that includes feature extraction, and part of the algorithm that includes the same feature extraction. This is somewhat confusing to me and maybe you want to review the structure of the sections.  You are telling us you are using the first layer (P=1) of the ResNet50 in the method description, and you mention that you are using the pre-final layer in the preprocessing section. I assume you are using the latter, or is P=1 identical to the prefinal layer in your notation?  Tell us. Moreover, not having read Durand 2016, I would appreciate a few more technical details or formal description here and there.  Can you detail about the ranking method in Durand 2016, for example?
* Would it make sense to discuss Durand 2016 in the base line methods section?
* To some degree this paper evaluates WELDON (Durand 2016) on new data, and compares it against and an extended WELDON algorithm called CHOWDER that features the final MLP step. Results in table 1 suggest that this leads to some 2-5% performance increase which is a nice result.  I would assume that experimental conditions (training data, preprocessing, optimization, size of ensemble) are kept constant in between those two comparisons? Or is there anything of relevance that also changed (like size of the ensemble, size of training data) because the WELDON results are essentially previously generated results? Please comment in case there are differences.

---

> ### Author Response · Authors · 2018-01-05
> **Response to Referee (Part 3 of 3)**
>
> > * Would it make sense to discuss Durand 2016 in the base line methods section?
>
> Considering the adaptations we make to the approach of Durand et al. (2016),
> we felt it more appropriate to cite their work within Sec. 2.3  in order to show the line of
> development. In Sec. 2.3, we cite Durand et al. (2016) extensively, pointing out the notable
> contributions of this work, and how the originally proposed approach must be adapted
> in order to provide an effective architecture for the setting of WSI classification
> without local annotations.
>
> In Sec. 2.2, when we introduce baseline techniques, we truly mean baseline. Aggregation
> via feature pooling is one of the most direct ways one can attempt to approach the
> task of WSI classification sans annotations. Indeed, this approach is very attractive,
> as compared to MIL approaches, when tackling large-scale datasets from a purely
> computational standpoint. For this reason, we denote these approaches as our "baseline,"
> whereby we demonstrate that either technique (WELDON or CHOWDER) can provide
> improvements in both detection and localization which are significant enough,
> as compared to feature pooling, to justify their complexity.
> By not including Durand et al. (2016) within Sec 2.2, we do not imply that we should not
> compare (we do), rather we simply make a semantic distinction between feature
> pooling and MIL.
>
> > * To some degree this paper evaluates WELDON (Durand 2016) on new data, and
> >compares it against and an extended WELDON algorithm called CHOWDER that features
> >the final MLP step. Results in table 1 suggest that this leads to some 2-5% performance
> >increase which is a nice result.  I would assume that experimental conditions (training data,
> >preprocessing, optimization, size of ensemble) are kept constant in between those two
> >comparisons? Or is there anything of relevance that also changed (like size of the ensemble,
> >size of training data)
>
> We do, in fact, evaluate the WELDON architecture of Durand et al. on new data,
> as reported in Table 1. We also compare WELDON against our proposed modifications.
> In all cases, experimental settings remain consistent between all tested methods. In the
> case of WELDON and CHOWDER, we have ensured that the ensemble size remains consistent
> between the two ($E = 10$ as described in Sec. 3.1). For both WELDON and CHOWDER,
> we use best-case hyper-parameter settings.
>
> Additionally, the improvement in AUC demonstrated by the CHOWDER architecture is more
> significant than the referee reports. In Table 1, we report a percent change in AUC
> of 12.15% and 8.53% over the WELDON architecture for the competition and
> cross-validation splits, respectively. This corresponds to a 12.59% and 23.66% percent
> change in AUC as compared to the best-performing baseline methods for the same splits.
> In the case of TCGA-Lung, we demonstrate a 1.32% percent change in AUC, but we point this
> out specifically in the text. This dataset is well suited to feature pooling due to
> the balanced instance classes present in the TCGA-Lung dataset.
> The diseased regions in these
> slides are much more diffuse over the entire tissue sample, as opposed to the
> highly-localized metastases present in Camelyon-16. Therefore, the excellent
> performance of the baseline feature-pooling methods is expected for this dataset,
> as the disease signal is not lost in the pooled representation.
>
> >because the WELDON results are essentially previously generated results?
> >Please comment in case there are differences.
>
> It is not clear to us what is meant by previously generated results. In the case of
> WELDON, in Durand et al. (2016), the method was proposed only for object region
> detection in natural images. To the best of our knowledge, there has been no other
> application of a WELDON-inspired architecture to HIA, or to the TCGA-Lung and
> Camelyon-16 datasets in particular.

---

> ### Author Response · Authors · 2018-01-05
> **Response to Referee (Part 2 of 3)**
>
> > * The intro starts from a very high clinical level. A introduction that points out
> >specifics of the technical aspects of this application, the remaining technical challenges,
> >and the contribution of this work might be appreciated by some of your readers.
>
> Since our work is focused on the application of machine learning techniques to a
> very significant and specific medical diagnostic, and given the very ML-focused audience
> of ICLR, we assume that the majority of readers will be unfamiliar with histopathological
> image analysis and clinical pathology in general. For this reason, we discuss the
> context of the problem at length, including the pernicious data challenges present in
> this application. Besides presenting our own contributions, we hope to introduce more
> researchers to this fruitful and important field. As modern machine learning techniques
> are only recently being applied to histopathological image analysis, especially in the
> weak-learning setting, there is a tremendous opportunity for interested readers to
> make a significant impact in this area.
>
> As for the presentation of our own contributions with respect to prior work, we have
> modified the last paragraph of the introduction to make these more clear.
>
>
> > * There is preprocessing that includes feature extraction, and part of the algorithm that
> >includes the same feature extraction. This is somewhat confusing to me and maybe you
> >want to review the structure of the sections.  You are telling us you are using the first layer
> >(P=1) of the ResNet50 in the method description, and you mention that you are using the
> >pre-final layer in the preprocessing section. I assume you are using the latter, or is P=1
> >identical to the prefinal layer in your notation?  Tell us.
>
> In our work, we propose the use of the ResNet-50 pre-output layer, namely, the
> values resulting from the convolutional stack, prior to the fully-connected output
> layers. We interpret these values as a feature vector describing the structural and
> color content of each $244\times 244$ pixel tile. In our notation, we use $P$ to refer
> to the dimensionality of this feature vector. In the case of ResNet-50, this layer has
> contains 2048 neurons, so we note that $P = 2048$. This use of an ImageNet pre-trained
> ResNet-50 architecture as a tile feature extractor remains consistent throughout the
> work, as described in the text. It is not clear to us at which point the referee
> found confusion between this definition of $P$ and ResNet-50
> layer indexing. If the referee would provide further details, we would be happy to
> clarify the text.
>
> As for the structure of the paper, we discuss the pre-processing stages prior to the
> introduction of both the feature-pooling techniques and CHOWDER since this pipeline
> remains consistent over all approaches.
>
> > Moreover, not having read Durand 2016, I would appreciate a few more technical details
> >or formal description here and there.  Can you detail about the ranking method in
> >Durand 2016, for example?
>
> By ranking method, we assume that the referee means the operation of the MinMax layer
> operating on the feature embedding values, as the modified ranking loss metric proposed
> in Sec. 4 of Durand et al. (2016) is easily substituted for binary cross-entropy loss
> in our binary classification setting, as we point out in Sec. 3.1.
>
> The instance ranking method used in Durand et al. (2016) during training, as well as in
> our approach, is simply sorting the embedding values in descending order,
> as we describe in the "Top Instances
> and Negative Evidence" subsection of Sec. 2.3. As for formal descriptions, we err on the
> side of brevity in light of the other necessary content in the paper. Since the
> application to HIA is novel to many readers, explanation and interpretation is
> required in order to relate the significance of our contribution. We have endeavoured to
> make details explicit when necessary to the presentation,
> such as tile selection and the operation of the feature embedding layer. In all other
> cases, we lean on the common expertise of the ICLR audience. If the referee would point
> us to any further ambiguities we would be happy to clarify the text.
>
> Additionally, the work of Durand et al. (2016) is very informative and
> presents an ingenious architecture. Given the referee's strong interest in our comparison
> to Durand et al. (2016), we would strongly encourage the referee to
> take the time to read their work in detail, as well.

---

> ### Author Response · Authors · 2018-01-05
> **Response to Referee (Part 1 of 3)**
>
> We thank the referee for their time in reviewing our work, we understand the significant
> time constraints placed upon many referees during this season. While we appreciate the
> call to a clear and defined statement of contributions and originality, we believe that
> perhaps the referee has misunderstood our work with respect to prior art. We hope to
> clarify the contributions of our work in our response, and with our modifications to
> the final paragraph of the introduction.
>
> > The authors approach the task of labeling histology images with just a single global label,
> >with promising results on two different data sets. This is of high relevance given the difficulty
> >in obtaining expert annotated data. At the same time the key elements of the presented approach
> >remain identical to those in a previous study ...
>
> We strongly contend this point. We assume that the referee is referring to the work of
> Durand et al. (2016) on the WELDON architecture. While we use this work as a starting
> point for our application, the specifics of our approach in the application to
> HIA are not at all identical to Durand et al. Indeed, the application to HIA
> (and the modifications required to achieve it)
> was not at all envisaged in the prior art, which was solely focused on object region
> detection in natural images (e.g. Pascal VOC, COCO, etc.). Indeed, the application to
> massive WSI datasets requires novel developments for acquiring instances during
> training (i.e. our proposed system of random sampling), much less the architectural and training
> changes we propose. Even the pre-trained DCNN is different from Durand et al. (2016),
> (ResNet-50 in place of VGG16). Given that there is no path from Durand et al. (2016)
> to human-level diagnosis prediction in WSI from diagnosis labels without the
> significant developments we outline, we cannot agree with the referee's assessment.
>
> >the main novelty is to replace the final step of the previous architecture
> >(that averages across a vector) with a multiplayer perceptron.  As such I feel that this
> >would be interesting to present if there is interest in the overall application
> >(and results of the 2016 CVPR paper), but not necessarily as a novel contribution to
> >MIL and histology image classification.
>
> As we detail in our general comments, we do indeed believe that there is a place at
> ICLR for works presenting state-of-the-art results for impactful applications, especially
> in medicine, and oncology diagnostics in particular. We also argue, as in our
> comments above, and in our general comments, for the novel contributions made by our
> work to MIL as well as to machine learning in HIA by presenting a human-level
> diagnosis prediction system for WSI trained without using disease annotation maps.

---

### Public Comment · (anonymous) · 2017-11-27
**Paper reproducibility**

Greetings to the authors of this paper,

Your paper is very interesting and insightful. As part of a reproducibility challenge, our team of students would like to attempt at reproducing the results of your paper. We are not affiliated with the official reviewers.

If it would be possible, it would be incredibly helpful if you are interested in providing parts of the code used in your implementations.

If you are interested, please comment below, and we can arrange to contact each other in private.

Thank you

---

### Author Response · Authors · 2018-01-05
**General Comments to Referees**

We thank the referees for their time and effort in reviewing our
work, especially in light of the many heavy review loads assigned this year.

We would like to address the general comments from the referees on the
significance of our work to the ICLR community. Specifically, whether or not the
CHOWDER architecture we propose for MIL in the context of histopathological image
analysis (HIA) represents both a reasonable contribution to the ever-growing body of MIL
research, as well as the suitability of such an application-specific paper to the
ICLR community at large.

In response to the first point, we make affirmative arguments for both our contribution
to the architecture of the proposed network, as well as for the procedural contribution,
demonstrating how to effectively regularize and train a network in the extreme
setting of weak learning at very small sample-to-feature ratios. In the case of
the proposed architecture, we believe that the additional multi-layer perceptron (MLP)
at the classification layer does represent a meaningful contribution to MIL research.
Specifically, the MLP allows a more context-aware bag classification from the
top- and bottom-ranked embedded instance representations.
In the case of Durand et al. (2016), a simple sum of these $2R$ values is used.
As reported in Durand et al. (2016), even this summation is itself an
incremental generalization of their "min+max" output reported in their MANTRA work
(Durand et al., 2015).

In the case of HIA, as evidenced by the results we report in Table 1, using the context
afforded by the distribution of embedded values *within* the top and bottom rankings
leads to significant improvements in AUC (a 12.15% and 8.53% percent change for the
competition and CV splits, respectively). This is especially critical for disease
detection in WSI, as diseased tissue is noted by its discrepancy from healthy tissue,
rather than an absolute description of a fixed feature set common to *all* diseased
tissue. Additionally, as in the case of metastasis detection in Camelyon-16, the
detection of highly localized regions (i.e. extreme instance class imbalance) requires
a more sensitive approach, such as that afforded by the use of an MLP, than the
embedding sum used in Durand et al. (2016). Therefore, we believe that
the results and method we report will be useful for readers seeking to train deep
nets in similar extreme MIL settings, especially those where class imbalance is a
noted concern.

Further, we note that the best-case AUC performance of an expert human pathologist for
Camelyon-16 was reported as 0.884 in Bejnordi et al. (2017), with the mean pathologist AUC
reported as 0.810. When using a large ensemble size, $E=50$, we report an AUC of 0.8706,
thus demonstrating diagnosis prediction performance better than the average human
 pathologist, but *without* making any use of expert assistance during training (e.g. disease
segmentation maps). This demonstrates that our proposed methodology is as effective as
an expert pathologist, while also allowing for machine ingenuity, as the model can adapt to
novel diseases and structures outside of the confines of human-produced segmentation maps.

In response to the second point, on the suitability of this work within ICLR,
we refer to the 2018 Call For Papers (CFP):

> We take a broad view of the field and include topics such as feature learning,
> [...] and issues regarding large scale learning...

> A non-exhaustive list of relevant topics:
>
> [...]
> * implementation issues, parallelization, software platforms, hardware
> * applications in vision, audio, speech, natural language processing, robotics,
>    neuroscience, or any other field...

Given that this work represents an application of machine learning techniques to
a very large-scale problem requiring novel contributions to an existing architecture,
and that we detail the many implementation issues required for the successful
utilization of our proposed approach, and that we provide state-of-the-art results
within a very relevant and socially impactful field, namely histopathological
image analysis as an oncology diagnostic, we believe that our work is in fact very
well suited and topical to ICLR.

---

### Author Response · Authors · 2018-01-05
**Paper Modifications**

We would like to point out the changes made to the text of our
submission to address the comments of the referees, as well as some of our own
corrections and clarifications.

1. We have updated the explanation for our choice of a univariate feature embedding ($J=1$)
versus a multivariate embedding ($J>1$) in light of more extensive
experimentation. Specifically, increasing the embedding dimensionality $J$ *can*
improve training loss. However, it diminishes generalization, providing
worse scores on held-out validation data. Even though our tested
datasets (Camelyon-16, TCGA-Lung) rival ImageNet in overall size after tiling,
the number of *unique* slide images remains very limited. In the weak-learning setting,
the training method may attempt to find any number of possible unique features that
could contribute to the overall WSI ("bag") class.
For binary WSI classification, restricting the model to
$J=1$ makes sense, as the positive/negative assignment of the embedding maps directly
to the binary classes (e.g. "contains cancer", "does not contain cancer"). When
setting $J>1$, this correspondence is lost, and it becomes much more difficult to both
regularize, as well as interpret, the model.

2. We have revised the references to a consistent format.

3. We have revised the metastasis detection figures (Figs. 4--6) with a more
legible/interpretable color map (*blue-white-red* from the previous *green-yellow-red*).

4. We have updated the last paragraph of the introduction to describe our
specific contributions with respect to prior work (Durand et al. 2016, in particular).

5. We have revised grammar, spelling, & usage in the text.

6. In the results section we have added references to the recently published work of
 Bejnordi et al. (2017), which reports human pathologist AUC performance on the
 Camelyon-16 dataset.

---

### Decision · Program_Chairs · 2018-01-29
**ICLR 2018 Conference Acceptance Decision**

**Decision:**

Reject

**Comment:**

Authors present a method for disease classification and localization in histopathology images. Standard image processing techniques are used to extract and normalize tiles of tissue, after which features are extracted from pertained networks. A 1-D convolutional filter is applied to the bag of features from the tiles (along the tile dimension, kernel filter size equal to dimensionality of feature vector). The max R and min R values are kept as input into a neural network for classification, and thresholding of these values provides localization for disease / non-disease.

Pro:
 - Potential to reduce annotation complexity of datasets while producing predictions and localization

Con:
- Results are not great. If anything, results re-affirm why strong annotations are necessary.
- Several reviewer concerns regarding novelty of proposed method. While authors have made clear the distinctions from prior art, the significance of those changes are debated.

Given the current pros/cons, the committee feels the paper is not ready for acceptance in its current form.